# Ectopic Rod Photoreceptor Development in Mice with Genetic Deficiency of WNT2B

**DOI:** 10.3390/cells12071033

**Published:** 2023-03-28

**Authors:** Alexandra K. Blomfield, Meenakshi Maurya, Kiran Bora, Madeline C. Pavlovich, Felix Yemanyi, Shuo Huang, Zhongjie Fu, Amy E. O’Connell, Jing Chen

**Affiliations:** 1Department of Ophthalmology, Boston Children’s Hospital, Harvard Medical School, 300 Longwood Avenue, Boston, MA 02115, USA; 2Division of Newborn Medicine, Boston Children’s Hospital, Boston, MA 02115, USA; 3Department of Pediatrics, Harvard Medical School, Boston, MA 02115, USA

**Keywords:** Wnt signaling, WNT2B, photoreceptor, retina, development

## Abstract

Wnt/β-catenin signaling is essential for embryonic eye development in both the anterior eye and retina. WNT2B, a ligand and activator of the Wnt/β-catenin pathway, assists in the development of the lens and peripheral regions of the eye. In humans *WNT2B* mutations are associated with coloboma and WNT2B may also assist in retinal progenitor cell differentiation in chicken, yet the potential role of WNT2B in retinal neuronal development is understudied. This study explored the effects of WNT2B on retinal neuronal and vascular formation using systemic *Wnt2b* knockout (KO) mice generated by crossing *Wnt2b*^flox/flox^ (fl/fl) mice with CMV-cre mice. *Wnt2b* KO eyes exhibited relatively normal anterior segments and retinal vasculature. Ectopic formation of rod photoreceptor cells in the subretinal space was observed in *Wnt2b* KO mice as early as one week postnatally and persisted through nine-month-old mice. Other retinal neuronal layers showed normal organization in both thickness and lamination, without detectable signs of retinal thinning. The presence of abnormal photoreceptor genesis was also observed in heterozygous *Wnt2b* mice, and occasionally in wild type mice with decreased *Wnt2b* expression levels. Expression of *Wnt2b* was found to be enriched in the retinal pigment epithelium compared with whole retina. Together these findings suggest that WNT2B is potentially involved in rod photoreceptor genesis during eye development; however, potential influence by a yet unknown genetic factor is also possible.

## 1. Introduction

Wnt/β-catenin signaling is a well-studied and well-conserved signaling pathway essential for embryonic eye development [1,2]. It regulates development in multiple cell types and regions of the eye, including the lens [3], retinal pigment epithelium (RPE) [4,5], and retinal vasculature [6], and is active throughout multiple developmental stages [1]. Wnt signaling acts within a large gene regulatory network in the embryonic eye, where it interacts with transcription factors and other signaling pathways, such as paired box 6 (Pax6), transforming growth factor beta (TGFβ), and bone morphogenetic proteins (BMP), to ensure proper tissue development [7,8,9].

WNT2B is a ligand and activator of the canonical Wnt/β-catenin pathway. Like other Wnt ligands, it binds to Wnt receptor complex Frizzled4/Lrp5/6 to subsequently stabilize β-catenin and allow for target gene transcription [10,11]. WNT2B has been shown to regulate the development of the peripheral regions of the eye [12,13] and to inhibit retinal neuronal differentiation in the embryonic chick eye [14]. It has also been reported that *Wnt2b* is highly expressed during BMP-induced RPE regeneration [7]. Humans with *WNT2B* mutations exhibit neonatal-onset chronic diarrhea [15], and a variety of ocular abnormalities, including iris colobomas and corneal defects, although it is unclear whether *WNT2B* is directly responsible for these ocular pathologies [16,17]. Overall WNT2B remains relatively understudied regarding retinal neuronal and vascular development.

In this study we investigated potential presence of ocular phenotypic abnormalities related to the previously described human ocular pathologies in mice with *Wnt2b* deficiency. We also explored the possible role of WNT2B in retinal neuronal and vascular development. We found that *Wnt2b*-deficient mice exhibit normal anterior segments and retinal vascular patterning. Yet a portion of *Wnt2b*-deficient mice displayed ectopic rod photoreceptor nuclei in the subretinal space, indicating abnormal rod maturation, differentiation, or migration. These findings suggest a potential involvement of WNT2B in photoreceptor genesis. Given that this phenotype was also occasionally observed in wild type (WT) mice as well at a lower frequency, potential influence or synergistic effects by another unknown gene cannot be excluded.

## 2. Methods

### 2.1. Generation of Wnt2b Knockout (KO) Mice

All animal studies were approved by the Boston Children’s Hospital Animal Care and Use Committee (protocol #00001722, approved 27 June 2022). The studies also adhered to the Association for Research in Vision and Ophthalmology Statement for the Use of Animals in Ophthalmic and Vision Research. 

Systemic *Wnt2b* KO (*Wnt2b*^−/−^) mice were generated by crossing *Wnt2b*^flox/flox^ (fl/fl) mice [18] in C57BL/6N background with CMV-cre mice (B6.C-Tg(CMV-cre)1Cgn/J, Stock number: 006054, Jackson Laboratory, in C57BL/6J background) to deplete *Wnt2b* (deletion of exons 2 and 3) in all tissues, including the germline. The founding *Wnt2b* heterozygous (het, *Wnt2b*^+/−^) breeding pairs were confirmed negative of *Crb1* (rd8) mutation (found in C57BL/6N background mice) to exclude its potential confounding effects [19], and were subsequently crossed for experimental litters. Both het and KO mice are viable and fertile. WT, fl/fl, and het were all examined as controls in different sets of experiments.

### 2.2. Eye Dissection and Examination of Anterior Segments and Lens

Appearance of the eyelid and anterior segment of the eye (including cornea, iris, and lens) was examined both in live adult mice and in dissected eyes, after euthanasia with CO_2_ inhalation followed by cervical dislocation. After eye enucleation, the lens was removed under a dissecting microscope by removing the anterior portion of the eye. Sizes of whole eyeballs and lens were measured using a ruler and analyzed in Adobe Photoshop. 

### 2.3. Analysis of Retinal Vasculature in Retinal Flat Mounts

Staining of retinal vasculature was performed following protocols as described previously [20,21]. Adult mouse eyes (>4 weeks old) were enucleated and fixed in 4% paraformaldehyde for 1 h. Retinas were dissected in 1X phosphate-buffered saline (PBS) under a dissecting microscope by removing the anterior portion of the eye and peeling away the choroid/sclera complex. Retinas were first permeabilized in cold 70% ethanol on ice for 30 min, followed by permeabilization in PBS with 1% Triton X-100 (PBSTx) for 1 h. Retinas were then incubated in isolectin GS-IB_4_ with Alexa Fluor 594 conjugate (Invitrogen, Waltham, MA, USA, Cat#: I21413) in 0.1% PBSTx overnight. Retinas were rinsed in PBS three times, then flattened onto slides by making 4 radial incisions with the photoreceptor side down. Slides were coverslipped using ProLong Glass Antifade Mountant (Invitrogen, Cat#: P36980) and imaged using a fluorescence microscope (AxioObserver.Z1 microscope; Carl Zeiss Microscopy, Oberkochen, Germany) to visualize vasculature in both superficial and deep layers.

### 2.4. Retinal Histology Characterization in Eye Cross Sections with H&E Staining

Retinal histology was examined in cross sections based on prior protocols [22]. After euthanasia, enucleated mouse eyes were embedded and frozen in Optimal Cutting Temperature compound (Sakura Finetek, Torrence, CA, USA, Cat#: 4583). Cross sections (12 µm) were cut using a cryostat (Leica Biosystems, Wetzlar, Germany) and placed on positively charged microscope slides (VWR, Radnor, PA, USA, Cat#: 16004-406). Sections were air-dried and briefly fixed in 4% paraformaldehyde for 15 min and stained with hematoxylin (Sigma-Aldrich, St. Louis, MO, USA, Cat#: HHS32-1L) for 3 min. Sections were then rinsed in ddH_2_O and developed in tap water for 5 min, followed by 10 dips in acid alcohol (0.5% hydrochloric acid in 70% ethanol) to remove excess stain. Sections were then stained with eosin Y solution (Sigma-Aldrich, St. Louis, MO, USA, Cat#: 1098441000) for 30 s and rinsed in ddH_2_O. The slides were dehydrated in a graded series of ethanol (50%, 70%, 95%, and 100%) and incubated with xylenes for at least 10 min. Slides were coverslipped using Permount Mounting Medium (Fisher Scientific, Waltham, MA, USA, Cat#: SP15-100) and air-dried before imaging.

### 2.5. Immunohistochemistry

Frozen cross sections were air-dried, then fixed in 4% paraformaldehyde for 15 min, followed by two times washing in PBS. Sections were then blocked in 5% goat serum (Sigma-Aldrich, St. Louis, MO, USA, Cat#: G9023-10ML) in PBS with 0.1% Triton X-100 for 1 h before incubation with primary antibody diluted in 5% serum in PBS overnight at 4 °C. Sections were washed in PBS before incubation with secondary antibodies and nuclear marker 4′,6-diamidino-2-phenylindole (DAPI) at room temperature for 1 h. Slides were coverslipped using Fluoro-Gel (EMS, Hatfield, PA, USA, Cat#: 17985-10) and imaged. Primary antibodies used: anti-RPE65 (Novus Biologicals, Centennial, CO, USA, Cat#: NB100-355SS), anti-rhodopsin (Millipore, Burlington, MA, USA, Cat#: MABN15), and anti-Ribeye (Synaptic systems, Goettingen, Germany, Cat#: 192103). Secondary antibodies used: Alexa Fluor 594 goat anti-mouse IgG (Invitrogen, Waltham, MA, USA, Cat#: A11032), and Alexa Fluor 488 goat anti-rabbit IgG (Invitrogen, Waltham, MA, USA, Cat#: A11034).

### 2.6. Fundus and OCT Imaging

Adult mice (2–4 months old) were anesthetized with ketamine/xylazine mixture, and their pupils were dilated with topical administration of Cyclomydril drops (Alcon Laboratories, Fort Worth, TX, USA). Fundus and optical coherence tomography (OCT) images were taken using a Micron IV imaging system (Phoenix Research Lab, Pleasanton, CA, USA) as previously described [22,23].

### 2.7. Polymerase Chain Reaction (PCR) and Genotyping

Genomic DNA was isolated from mouse tails by first lysing tails in 25 mM NaOH/0.2 mM EDTA at 98 °C for 1 h and then adding 40 mM Tris HCl and centrifuging. *Wnt2b* and *rd8* PCR was performed using OneTaq Quick-Load 2X master mix (New England BioLabs, Ipswich, MA, USA, Cat#: M0486S) and followed protocols described previously [18,19,24]. For cre genotyping, PCR was performed with Apex 2X Taq RED master mix (Genesee Scientific, San Diego, CA, USA, Cat#: 42-138). Amplicons were separated using a 1% Tris-acetate-EDTA agarose gel stained with SYBR Safe (ThermoFisher, Waltham, MA, USA, Cat#: S33102) and visualized using an Azure 600 imaging system (Azure Biosystems, Dublin, CA, USA).

The following primers and protocols were used for PCR genotyping.

Genotyping for *Wnt2b*:

*Wnt2b* primers [18]:

KO-1: GCCTCTCACACCAGCGTGTAAGAG; 

KO-4: GTAATTGAGTGGTCTCCACC; 

2BNEO: ATCAGCAGCCTCTGTTCCACATAC.

*Wnt2b* Reactions were initially incubated at 94 °C for 2 min. Reactions were then denatured at 92 °C for 15 s, annealed at 70 °C for 30 s, and elongated at 68 °C for 2.5 min. These steps were repeated for 9 cycles with the annealing temperature decreasing by one degree every cycle. Reactions then underwent 30 cycles at 92 °C for 15 s, 60 °C for 30 s, and 68 °C for 2.5 min, with a final incubation at 68 °C for 10 min.

Amplicon sizes for *Wnt2b* WT: 200 bp, KO: 320 bp. 

Genotyping for *CMV-Cre:*

CMV-Cre primers: 

oIMR1084: GCG GTC TGG CAG TAA AAA CTA TC;

oIMR1085: GTG AAA CAG CAT TGC TGT CAC TT;

oIMR7338: CTA GGC CAC AGA ATT GAA AGA TCT (Internal positive control);

oIMR7339: GTA GGT GGA AAT TCT AGC ATC ATC C (Internal positive control).

Cre reactions were initially incubated at 95 °C for 4 min. Reactions were then continued at 95 °C for 30 s, annealed at 58 °C for 30 s, and elongated at 72 °C for 30 s. These steps were repeated for 35 cycles, with a final incubation at 72 °C for 7 min.

Amplicon sizes for *Cre*: 100 bp.

Genotyping for *Crb1* (*rd8*) [19]:

m*Crb1* mF1: GTGAAGACAGCTACAGTTCTGATC; 

m*Crb1* mF2: GCCCCTGTTTGCATGGAGGAAACTTGGAAGACAGCTACAGTTCTTCTG; 

m*Crb1* mR: GCCCCATTTGCACACTGATGAC.

Rd8 reactions were initially denatured at 94 °C for 5 min followed by 35 cycles at 94 °C for 30 s, 65 °C for 30 s, 72 °C for 30 s, and a final extension at 72 °C for 7 min.

Amplicon sizes for *Crb1* (*rd8*) are WT: 200 bp, rd8: 244 bp. 

C57BL/6J (from Jackson Lab, Bar Harbor, ME, USA, stock # 000664, known to be absent of rd8) mouse tail samples were used as WT control for *rd8* genotyping.

### 2.8. RNA Isolation and Real-Time Quantitative PCR (RT-qPCR)

RT-qPCR experiments were carried out based on previous protocols [25]. Tissue samples from *Wnt2b* mice were first homogenized and lysed using QIAzol Lysis Reagent (Qiagen, Germantown, MD, USA, Cat#: 79306). Total RNA was isolated using PureLink RNA Mini Kit (Ambion, Austin, TX, USA, Cat#: 12183025) following manufacturer protocols. RNA concentration and quality was tested using a NanoDrop 8000 (Thermo Scientific, Waltham, MA, USA). RNA was further purified using DNase I treatment (ThermoFisher, Waltham, MA, USA Cat#: EN0521) following manufacturer protocols. cDNA was synthesized using iScript Reverse Transcription Supermix (Bio-Rad, Hercules, CA, USA, Cat#: 1708841) following manufacturer protocols. RT-qPCR was performed using a CFX96 Real-Time System (Bio-Rad, Hercules, CA, USA) with SYBR Green qPCR Master Mix (ThermoFisher, Cat Waltham, MA, USA #: 4309155). The primer sequences of target genes are listed in Table 1. Copy number of each target gene cDNA was normalized to the housekeeping gene, *Gapdh*, using the comparative CT (ΔΔCT) method.

### 2.9. Statistical Analysis

Quantitative data are presented as mean ± SD. *p* values listed were calculated using multiple Student’s *t* tests for two groups of data. *p* values < 0.5 were considered statistically significant. 

## 3. Results

### 3.1. Normal Appearance of Anterior Segments of Wnt2b KO Mice

Genetic loss of *Wnt2b* in KO mice was validated with tail snip genotyping and RT-qPCR. We first examined the anterior segments of the *Wnt2b* KO mice to determine if any similar morphological abnormalities existed between this mouse strain and the previously reported human ocular phenotypes [16]. Grossly normal appearance of eyelids and cornea with clear lens was observed in both WT and *Wnt2b* KO mice, with comparable eye size and lens diameter, as well as relatively normal iris size and shape (Figure 1A–E). These results suggest that *Wnt2b* KO mice did not appear to reproduce the human ocular phenotype with anterior segment defects associated with *WNT2B* mutation.

### 3.2. The Retinal Vasculature of Wnt2b KO Mice Appears Normal

WNT2B binds Frizzled4 (Fzd4) [13], and Fzd4 signaling is linked with familial exudative vitreoretinopathy in humans and in mice with incomplete development of retinal vasculature [26]. We therefore examined the retinal vasculature of *Wnt2b* KO mice. Using retinal flat mounts with isolectin staining, we found that WT, *Wnt2b* Het, and KO adult mice all showed grossly normal retinal vascular patterning in superficial and deep vessel layers in flat mounts (Figure 2A–F), which is also confirmed in eye cross sections (Figure 2G–I). Together these findings suggest that WNT2B is dispensable for retinal vascular development and not required for activating Frizzled4 signaling to exert its role in retinal angiogenesis. 

### 3.3. Ectopic Formation of Rod Photoreceptor Clusters in the Subretinal Space in Adult Wnt2b Mutant Mice

Because of presence of abnormal DAPI-positive cell clusters in *Wnt2b* KO retinas (Figure 2F), we next performed H&E staining on retinal cross sections to fully investigate retinal neuronal development and lamination. In adult mice (2–5 months old), we found clusters of ectopic cells in the subretinal space, near the photoreceptor inner and outer segments and RPE, in both *Wnt2b* Het and KO mice (Figure 3A–C). Localization of these ectopic cells is random across the retina. All other retinal neuronal layers appeared normal in both thickness and organization (Figure 3A–C). The *Wnt2b* fl/fl control mice, on the other hand, did not exhibit this phenotype (Figure 3A).

To identify the nature of these ectopic cells, we performed immunohistochemistry with photoreceptor cell markers. The ectopic cells in *Wnt2b* Het and KO retinas showed positive staining for rhodopsin (Figure 3D–F). In addition, staining for Ribeye, a structural protein of ribbon synapse characteristic of photoreceptor synapses, was also positive around a group of ectopic cells that extended aberrantly from the normal outer nuclear layer in KO retinas (Figure 3G–I). These findings indicate that these abnormal cells are displaced mature rod photoreceptor cells with photo-sensing pigments and ribbon synapse.

### 3.4. Ectopic Subretinal Cell Clusters Form during Wnt2b Mutant Mice Development

To elucidate the timing of the onset of this phenotype, we examined mice at developmental time points P7, P15, and P30. Ectopic cells were present in both *Wnt2b* Het and KO mice at all three of these time points (Figure 4A–C), suggesting that the malformation likely occurs shortly after birth during the retinal progenitor cell differentiation process. At P7 and P15 only, we also observed pigment near the ectopic cells, primarily within the displaced inner and outer segments in KO retinas (Figure 4C); however, these pigmented structures were not colocalized with cell nuclei marker DAPI (Figure 4D), indicating an acellular nature or cell debris following loss of nuclei.

In addition, the samples with displaced pigments (P15) did not show positive staining for RPE65, an enzyme and marker specific to RPE cells (Figure 4E,F). This suggests these pigments are not functioning RPE cells.

### 3.5. Absence of Retinal Thinning and Degeneration in Aged Wnt2b KO Mice

To determine whether the ectopic cells may spontaneously regress over time and whether potential aging effects may alter or confound the observed phenotype, we evaluated 9-month-old KO retinas. Although the frequency of the phenotype at 9 month is lower (1/4 of Het and KO mice combined, Table 2) compared with younger ages in frequency analysis (Table 2), we did observe ectopic cells in a 9-month-old *Wnt2b* KO retina (Figure 4G,H), confirming that this phenotype can persist well into aging. Except for disorganization of the photoreceptor layer due to the misplaced photoreceptor cells, we did not detect obvious signs of retinal thinning or other forms of age-related retinal degeneration in *Wnt2b* KO mice when compared to WT mice at 9 months old (Figure 4G,H), suggesting that the phenotype is mostly a developmental defect that persists into adulthood yet does not appear to progress into retinal thinning and degeneration in later age.

### 3.6. Lack of Phenotype Detection in Fundus Imaging and OCT in Live Mice

In addition to morphological characterizations in dissected eyes, we performed fundus and optical coherence tomography (OCT) imaging on live *Wnt2b* mutants to determine if this phenotype could be observed on a macroscopic scale to facilitate live screening and monitoring. We did not observe any obvious abnormalities in the fundus images of *Wnt2b* Het or KO mice when compared to WT (Figure 5A,B). We were also unable to detect any gross variations in retinal structure or lamination in the OCT images (Figure 5C,D). In this technique, the light path is confined to the central retinal area and peripheral regions are not visible in the fundus imaging, potentially limiting our ability to evaluate the full retina through fundus imaging, even when the phenotype was confirmed in retinal sections (Figure 5E,F).

### 3.7. Occasional Presence of Phenotype in WT Mice

Despite our initial results suggesting global loss of *Wnt2b* may be the direct cause of this novel phenotype, we later observed the same phenotype in a small number of littermate WT control mice. Although the frequency was much lower in WT or fl/fl mice (17%) than in Het (53%) or KO (35%) mice (Table 2), we found similar clusters of displaced cells in WT retinal cross sections at P15, P30, and 4 months old (Table 2, Figure 6A–C). In P15 retinas, the ectopic displaced cells are also associated with pigment deposition (Figure 6A).

To confirm their genotype, we performed RT-qPCR on eyecup samples from mice, of which one WT had shown ectopic cells in cross sections. Our results were consistent with the previous genotype results from tail snips, with the abnormal phenotypic WT showing consistent *Wnt2b* expression at a lower level than, but statistically insignificant from, the WT without the phenotype (Figure 6D). Absence of *Wnt2b* expression in the KO eye was also confirmed with RT-qPCR, as expected (Figure 6D). This observation in the WT suggests two possibilities: WT eyes may express varying levels of *Wnt2b*, and those with lower levels may manifest a phenotype similar to that observed in Het and KO mice; or an unknown gene mutation may cause the displaced photoreceptor phenotype independently of or in combination with loss of *Wnt2b*.

### 3.8. Expression of Wnt2b mRNA in the RPE and Retina

To understand the ocular cellular expression of *Wnt2b*, expression of *Wnt2b* mRNA in RPE and whole retina from adult eyes was compared (Figure 7A), which showed enriched levels of *Wnt2b* in mouse RPE as compared to the retinas. Relative expression of *Wnt2b* and its potential Wnt receptors (FZD4, FZD7) and coreceptors (LRP5 and LRP6) in retinal cells were analyzed with published single-cell sequencing data from P14 mice [27] (Figure 7B). Within the retina, *Wnt2b* showed modest expression throughout retinal cells including photoreceptors (rods and cones), bipolar cells, and ganglion cells, with the exception of cholinergic amacrine cells, which have higher levels and more enrichment of *Wnt2b* than other cell types (Figure 7B).

### 3.9. Expression of Wnt Ligands and Frizzled (Fzd) Receptors Was Altered in Wnt2b KO Retinas

To assess whether other Wnt ligands and Fzd receptors are regulated in the absence of *Wnt2b*, we evaluated their mRNA expression levels in WT and *Wnt2b* KO retinas. Expression of *Wnt2b* was not detectable in KO retinas, as expected (Figure 8A). Relative expressions of *Wnt2*, *Wnt4*, and *Wnt5b* were down regulated, whereas *Wnt10a* was modestly upregulated in *Wnt2b* KO retinas (Figure 8A). Levels of *Wnt3a*, *Wnt5a*, and *Wnt7a* were comparable (Figure 8B). For the receptors, we found that expression levels of *Fzd2* were significantly upregulated and *Fzd8* were downregulated in Wnt2b KO retinas (Figure 8C), whereas *Fzd1*, *Fzd3*, *Fzd4*, *Fzd6*, and *Fzd7* did not show significant change (Figure 8D). These data suggest that altered expression of Wnt ligands and receptors may potentially compensate for the loss of function of WNT2B in its absence.

In addition, we also analyzed the expression of neurogenin1. Overexpression of neurogenin1 was found to cause ectopic retinas with multiple ectopic cell types and layers in the subretinal space [28,29,30], resembling somewhat the phenotype observed in *Wnt2b* KO eyes. We found that expression of neurogenin1 showed a trend of upregulation in *Wnt2b* KO retinas compared with WT, although it was statistically insignificant (Figure 8E). This suggests a possibility that neurogenin1 dysregulation may potentially explain in part the phenotype in *Wnt2b* KO eyes.

## 4. Discussion

In this study, we examined mice with global genetic deficiency of *Wnt2b* to determine if they exhibit any ocular abnormalities and thus could be a useful mouse model to study *WNT2B*-related ocular birth defects found in humans [16,17]. In contrast to the findings in humans where anterior segment abnormalities were linked with *WNT2B* mutations, in *Wnt2b* KO mice we did not observe detectable gross abnormalities of the anterior eye, suggesting that *Wnt2b* is likely dispensable in anterior eye structure formation, and has a distinctly different role in mice versus humans. Because Fzd4, a major receptor for WNT2B, is important for retinal angiogenesis [26], we also examined formation of retinal vasculature in *Wnt2b* KO eyes. We found normal development of retinal vessels and patterning, suggesting that WNT2B is not requried for retinal blood vessel growth and formation in mice. 

Importantly, we found abnormal retinal neuronal genesis with ectopic clusters of cell nuclei aggregated in the subretinal space. These cells appeared to be displaced aberrantly from the outer nuclear layer. We further confirmed these cells were of rod photoreceptor origin with ribbon synapse expressing both photoreceptor and synaptic markers. These abnormally displaced nuclei were present at P7 through adulthood to 9 months of age. During retinal development at P7 and P15, these nuclei were also accompanied by ectopic pigments of unknown origin, which disappeared at later time points. In addition, we found *Wnt2b* expression is higher in RPE than in the retina. Together, this observed phenotype of ectopic nuclei may reflect misguided photoreceptor differentiation and migration during development, potentially in the absence or knockdown of *Wnt2b*. This notion is consistent with the known role of β-catenin/Wnt signaling in controlling retinal progenitor cell fate and in photoreceptor development and regeneration. For example in chick eyes, *Wnt2b* expression was found in the RPE and the peripheral retina to inhibit differentiation of retinal neurons and regulate peripheral eye formation [12,13]. 

One interpretation is that disruption of WNT2B in mice primarily affects the RPE cells that typically guide rod photoreceptor development and lamination, causing this process to be disrupted and for additional, ectopic rod cells to develop and/or migrate incorrectly [31]. This notion is supported by our findings of enriched *Wnt2b* expression in the mouse RPE/choroid/sclera complex, which was significantly higher when compared to the retina (Figure 7). Expression levels of *Wnt2b* in many retinal cells are modest, except for in cholinergic amacrine cells (Figure 7). Receptors for *Wnt2b*, including frizzled4 (fzd4), fzd7 [32], and LRP5/6, are also expressed in the retina (Figure 8), and may potentially mediate the signaling of WNT2B. If WNT2B is at least partially responsible for the phenotype observed, WNT2B secretion from the RPE may thus guide retinal neuronal specification or differentiation through these receptors. 

Alternatively, it may be possible that some RPE cells may transdifferentiate into ectopic retinal neuronal cells. Wnt signaling is known to be essential for early eye and RPE development, through interaction with other factors such as Mitf, Otx2, and BMP, and previous work shows that Wnt inactivation results in RPE transdifferentiation to neural retinal cells in embryonic mice [5,33]. Given that we observed disrupted lamination as early as P7, altered *Wnt2b* expression and potential dysregulation of other related developmental factors could result in RPE differentiation into ectopic rod photoreceptor cells. Particularly, Pax6 and Mitf are known to be connected to both Wnt signaling and RPE differentiation [34]. 

The fact that the observed ocular phenotype was also occasionally present in WT mice also suggests the alternative possibility of an unknown factor or mutation present in this strain that may either directly cause the observed phenotype independently of *Wnt2b*, or may interact with *Wnt2b* to precipitate the ocular phenotype. In this regard, one possible mutation we ruled out was the *rd8* allele of the *Crb1* gene. *Rd8* mice show a somewhat similar disruption of retinal nuclear layers, accompanied by retinal degeneration and abnormal fundus imaging with spotting [19]. It was previously discovered that C57Bl/6N mice can harbor this mutation [19], which can confound interpretations of ocular phenotypes. The original *Wnt2b* fl/fl mice were of a C57Bl/6N background [18], and were crossed to CMV-Cre mice of C57Bl/6J background. We confirmed with genotyping that the *Wnt2b* KO mice lack the *rd8* allele. Moreover, *Wnt2b* fl/fl mice did not show the abnormal ocular phenotype as observed in *Wnt2b* KO mice, suggesting that the observed phenotype is unrelated to *rd8*. 

Another potential source of confounding genetic influence may come from the CMV-Cre breeder mice. The *Wnt2b* KO colony was not selectively bred to remove the Cre gene, hence the CMV-Cre gene was still present throughout the colony. Since all mice in this colony still contained CMV-Cre, the Cre recombinase gene itself is unlikely to be responsible the ocular phenotype. However, it is possible that an unknown gene originated from CMV-Cre strain may contribute to the phenotype independently or may interact with WNT2B to alter its expression or function and thereby leading to the observed ocular phenotype. Indeed, one out of six CMV-Cre eyes examined exhibited a similar phenotype of ectopic photoreceptors, suggesting the possibility of an unknown genetic factor originating from CMV-Cre strain background. To this end, it is worth noting that a separate strain of systemic *Wnt2b* KO mice was generated using the same *Wnt2b* fl/fl mice but bred with actin-Cre mice, another systemic Cre strain [18], where no gross eye abnormalities in external appearance in *Wnt2b* KO mice were reported. However, it is unclear in that study whether detailed examination of eye anatomy and retinal cellular structure was performed.

Although our observed ocular phenotype shows some similarities to those of certain known mutations, no previously discovered ocular phenotype truly matches what we have described in this study. Most observations of disrupted lamination and displaced photoreceptor nuclei in mice appear to result in significant retinal thinning and degeneration with age. The rd6 mutation of the Crb1 gene, for instance, causes ectopic, pigmented nuclei to develop in the subretinal space, but this mutation is also accompanied by severe retinal degeneration and spotting on fundus images [35]. Similarly, the microphthalmia mutation (loss of Mitf) in mice causes disrupted retinal neuronal patterning due to dysfunctional RPE differentiation, but is again accompanied by severe retinal degeneration as well as grossly small eye size [36], which were not observed in our mice. More similar to what we observed here was the phenotype seen in mice with expression of constitutively active RAC1 in rod photoreceptors [37]. These mice displayed abnormal retinal neuronal lamination accompanied by incorrect photoreceptor polarity, leading to improper migration during development. Although the histology at developmental time points closely matches our results, mice with RAC1 expression also showed significant photoreceptor loss by 3 months. 

The closest phenotype reported thus far was seen in mice with overexpression of *neurogenin1*, a proneural gene. These mice developed an ectopic retina in the subretinal space with multiple cell types and layers [28,29,30]. Although the phenotype we observed was less severe and spontaneous, neurogenin1 might be a promising candidate in explaining the ectopic cells in our mice, which is consistent with our findings of a trend of *neurogenin1* upregulation in *Wnt2b* KO retinas (Figure 8). Interestingly, the Drosophila homolog of neurogenin, Tap, was also found to interact with Wnt protein Dishevelled to direct neuronal development [38]. Thus, it is possible that *Wnt2b* mutation might be linked with other neuronal guidance factors such as *neurogenin1*, which can cause a similar phenotype. A potential separate independent mutation of these other factors such as *neurogenin1* also cannot be ruled out.

One significant technical aspect that limited our study is that we were only able to observe the ocular phenotype in retinal cross sections ex vivo, but not in live mice with fundus imaging or OCT. This made it technically challenging and impractical to screen a large number of mice to select those affected mice to further evaluate their visual function and other characteristics. Considering the microscopic nature and limited regions with the abnormal photoreceptors, it is possible that visual function might not be severely impaired in those affected mice. 

To summarize, in this study we observed ectopic rod photoreceptor nuclei unaccompanied by retinal degeneration in a strain of systemic *Wnt2b* deficient mice. Given that this was also observed in a limited number of WT mice, we speculate that some other factor, possibly in conjunction with altered *Wnt2b* expression, is potentially responsible for this phenotype. More in-depth analysis will be needed to fully understand the nature of the genetic causes of the observed ocular phenotype. Nevertheless, the presence of ectopic photoreceptor genesis can make these mice a useful model to study photoreceptor differentiation and migration, with potential implications for photoreceptor regeneration as well.

## Figures and Tables

**Figure 1 cells-12-01033-f001:**
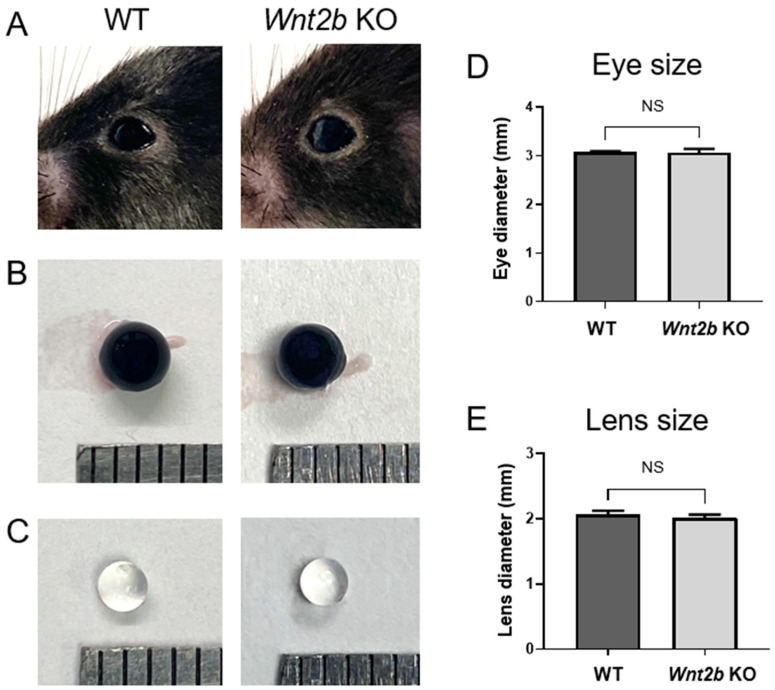
**Wnt2b KO mice have normal anterior segments with comparable eye and lens sizes.** (**A**–**C**) Representative images of mouse eyes’ exteriors (**A**), isolated whole eyeballs (**B**) and lens (**C**) from 1-month-old WT and *Wnt2b* KO mice. (**D**,**E**) Quantification of the diameters of isolated whole eyeballs (**D**) and lens (**E**). Scale: each grid is 1 mm. Data presented as mean ± SD; n = 4–8 eyes/group; NS: not significant.

**Figure 2 cells-12-01033-f002:**
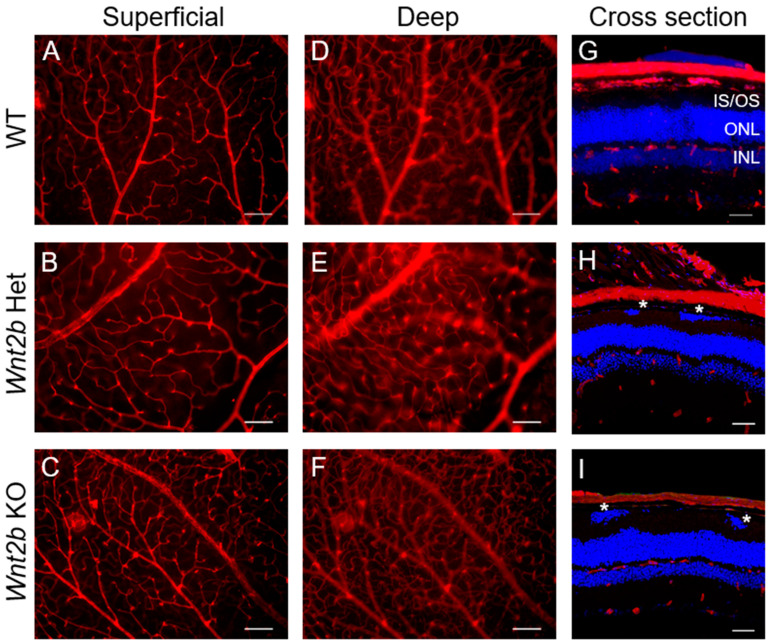
***Wnt2b* KO mice exhibit normal retinal vasculature.** (**A**–**F**) Retinal flat mounts stained with isolectin (red) showing normal vasculature of young adult WT (**A**,**D**) *Wnt2b* Het (**B**,**E**) and KO (**C**,**F**) mice. Images (**A**–**C**) show superficial layer and (**D**–**F**) show deep layer vessels. Scale bars: 100 µm. (**G**–**I**) Retinal cross sections from young adult mice were stained with isolectin (red) and counterstained with DAPI (blue) and showed fully formed and normal vascular layers (superficial, intermediate, and deep). Scale bars: 50 µm. * indicates ectopic nuclei in IS/OS. IS/OS: inner segments/outer segments; ONL: outer nuclear layer; INL: inner nuclear layer.

**Figure 3 cells-12-01033-f003:**
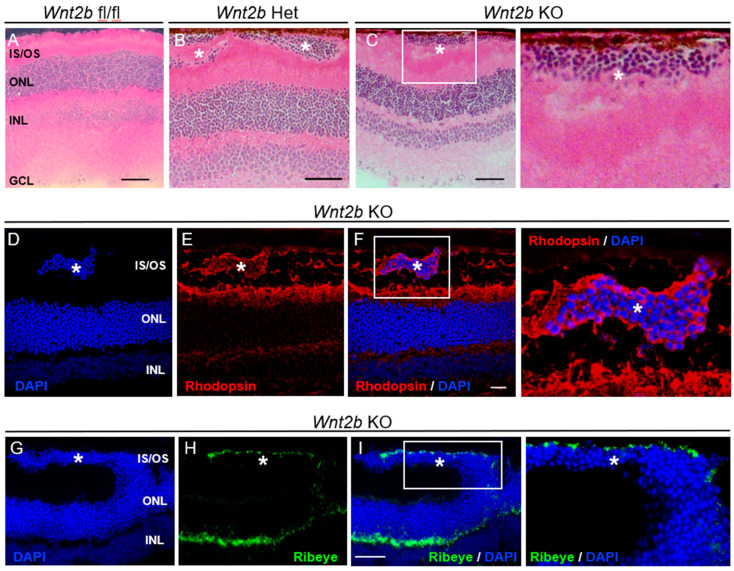
**Ectopic rod photoreceptor nuclei found in *Wnt2B* Het and KO mice.** (**A**–**C**) H&E staining on retinal cross sections from 2–3-month-old *Wnt2b* fl/fl (**A**), Het (**B**) and KO (**C**) mice. Ectopic nuclei can be seen in the subretinal space near RPE in Het and KO , but not fl/fl mice. * indicates ectopic nuclei in the subretinal space. Scale bars: 50 µm. An area with ectopic nuclei was enlarged on the right in the KO image (**C**). (**D**–**I**) Immunohistochemistry on retinal cross sections from 2-month-old *Wnt2b* KO eyes, stained for rhodopsin (red, **E**,**F**), Ribeye (green, **H**,**I**), and co-stained with DAPI (blue, **D**,**F**,**G**,**I**). Ectopic cells (*) stained positively for rhodopsin and Ribeye, indicating their rod photoreceptor origin with ribbon synapse. Areas with ectopic nuclei were enlarged on the right in the overlay image (**F**,**I**). Scale bars: 20 µm (**F**), 50 µm (**I**). IS/OS: inner/outer segments; ONL: outer nuclear layer; INL: inner nuclear layer; GCL: ganglion cell layer.

**Figure 4 cells-12-01033-f004:**
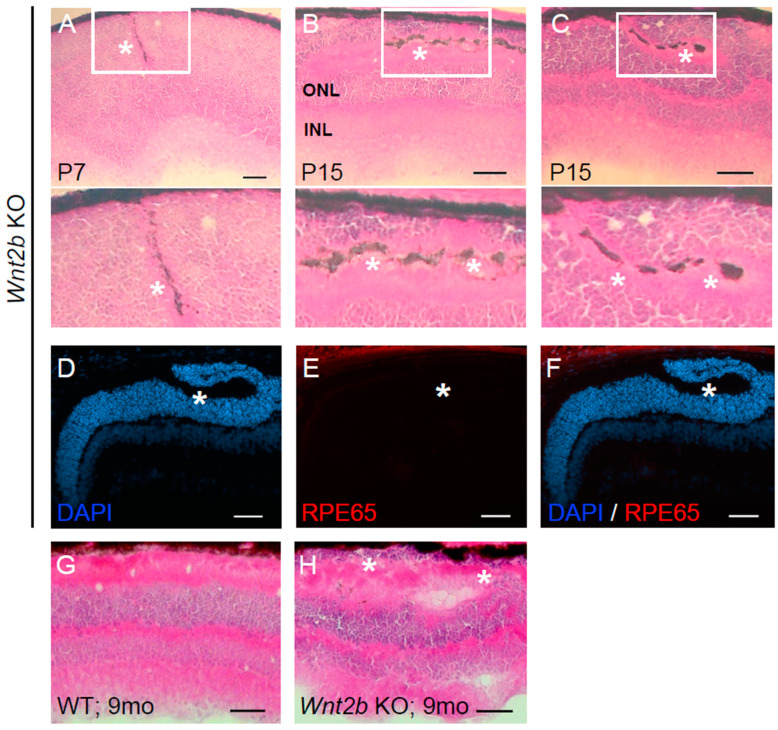
**Abnormalities in retinal lamination begin as early as P7 and persist through 9 months old in *Wnt2b* KO eyes.** (**A**) P7 and (**B**,**C**) P15 *Wnt2b* KO retinal cross sections with H&E staining showing ectopic nuclei and disrupted retinal neuronal layers, with abnormal pigmentation (*) around the ectopic cells. Selected areas around the pigmentation were enlarged in the lower portion of panels. (**D**–**F**) Immunohistochemistry of a retinal section adjacent to the tissue section as shown in (**C**), stained for RPE cell marker RPE65 (red), and co-stained with DAPI (blue). * indicates the position of ectopic pigments negative for RPE65, indicating non-RPE origin or potentially regressing and non-functional RPE without RPE65. (**G**,**H**) H&E staining from 9-month-old WT (**G**) and *Wnt2b* KO (**H**) eye cross sections. Ectopic cells (*) are visible in the KO sample. No evidencse of significant retinal thinning was found at this time point. Scale bars (**A**–**H**): 50 µm.

**Figure 5 cells-12-01033-f005:**
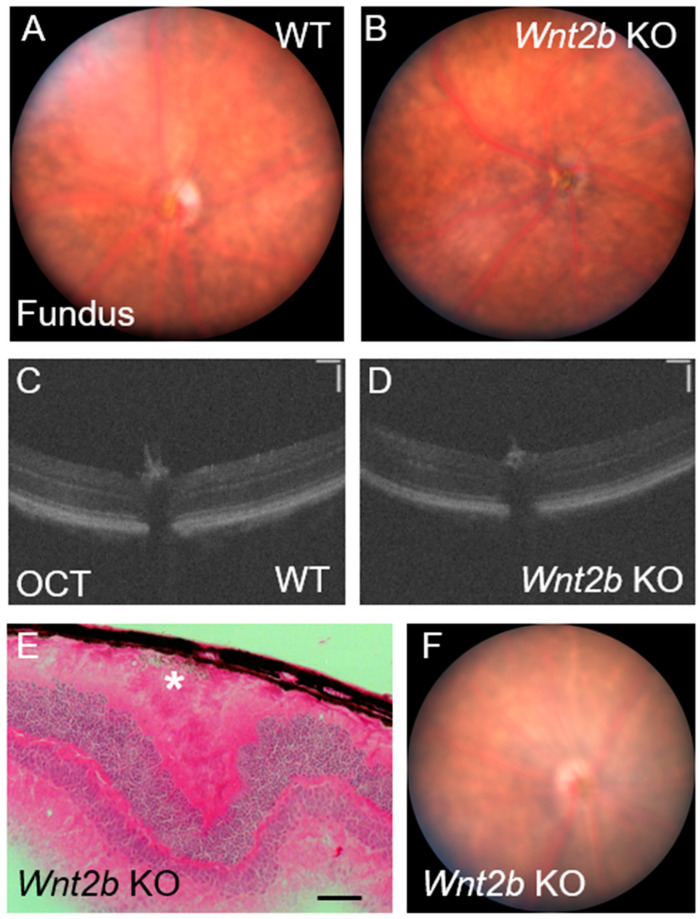
**In vivo fundus and OCT imaging does not reveal abnormalities in *Wnt2b* KO mice.** (**A**,**B**) Fundus images from 4-month-old WT (**A**) and *Wnt2b* KO (**B**) mice. No obvious retinal spotting or abnormalities can be observed. n= 4 mice/group (n = 2 at P30, n = 2 at 4 months). (**C**,**D**) OCT images from 2-month-old WT (**C**) and *Wnt2b* (**D**) KO mice. No obvious disruptions to retinal structure or lamination can be detected. n = 2/group. Scale bars: 100 µm. (**E**,**F**) H&E staining (**E**) of P30 *Wnt2b* KO mice, showing the rosette-like formation and ectopic cells (*), and fundus image (**F**) for the same eye show absence of any obvious retinal spotting, indicating this phenotype is unlikely to be detected with in vivo imaging. Scale bar: 50 µm.

**Figure 6 cells-12-01033-f006:**
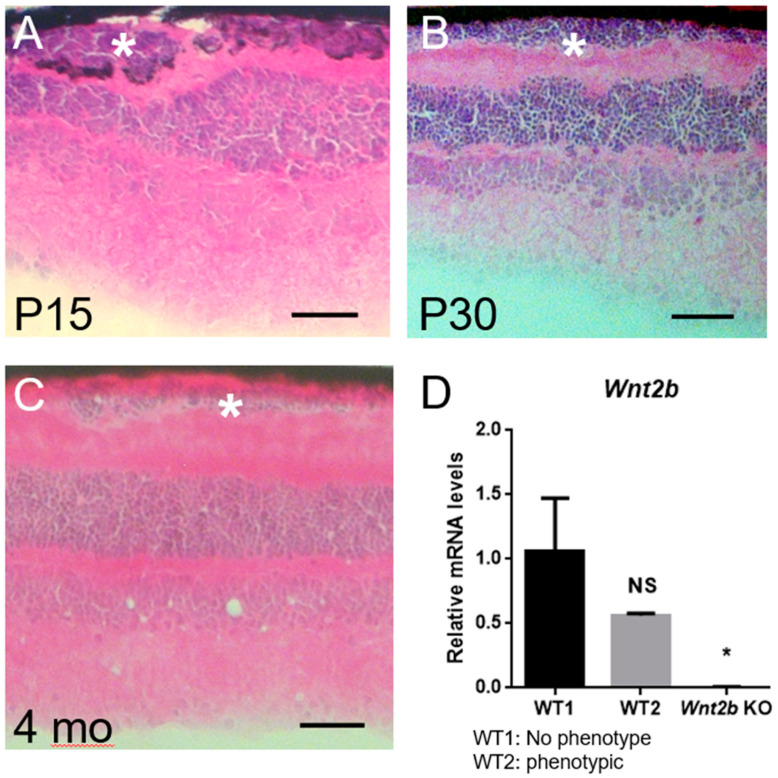
**WT eyes occasionally show phenotype of ectopic photoreceptors, indicating the possibility of altered *Wnt2b* expression or unknown alternative genes involved.** (**A**–**C**) H&E staining on WT retinal cross sections at P15 (**A**), P30 (**B**), and 4 months old (**C**) showing ectopic nuclei (*). Scale bars: 50 µm. Image (**A**) also shows ectopic pigments, consistent with pigment observation in KO samples at P15 (shown in Figure 4B,C). (**D**) qPCR of *Wnt2b* mRNA expression in WT1 (without phenotype), WT2 (with phenotype), and KO mouse eyes from age-matched adult mice (4 months old). Lack of *Wnt2b* expression is confirmed in KO, and the phenotypic WT2 mouse shows lower yet statistically insignificant levels of *Wnt2b* compared with WT1 without phenotype. Each sample contains a half eye cup with retinas after lens removal, with histological analysis of phenotype performed in the other half of the same eye for confirmation. For RT-qPCR, n = 3 technical repeats/group. Data presented as mean ± SD. * *p* < 0.05 KO compared to WT1. NS: not significant.

**Figure 7 cells-12-01033-f007:**
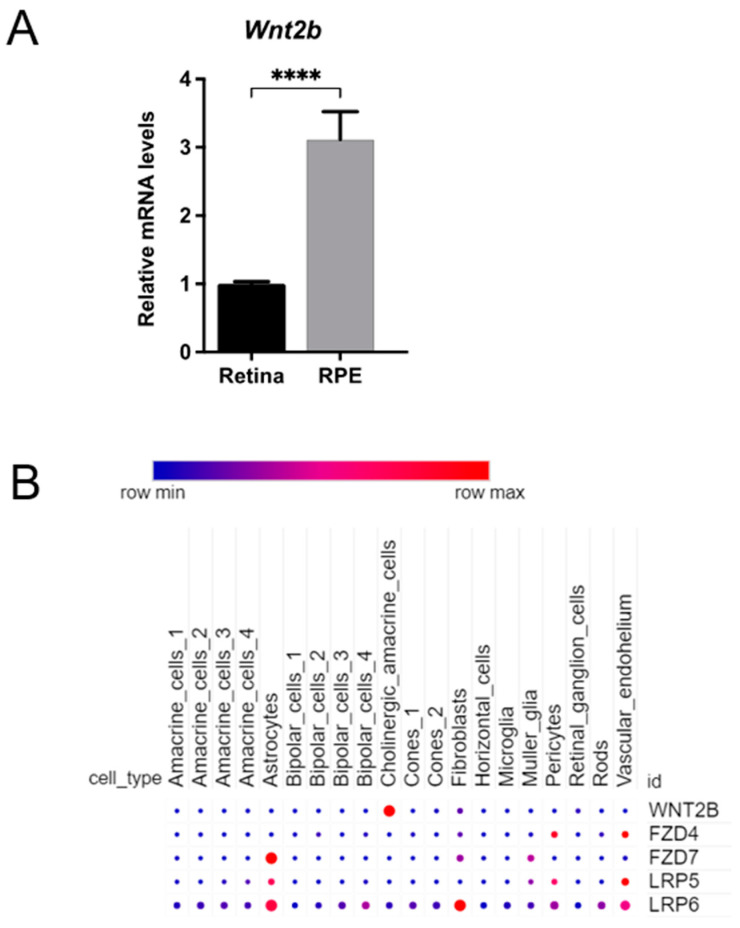
**Expression of *Wnt2b* in mouse RPE and retina.** (**A**) Relative expression of *Wnt2b* in RPE versus whole retina in adult C57BL/6J mouse eyes, suggesting enrichment of *Wnt2b* in RPE. n = 3–4 eyes /group. **** *p* < 0.0001. (**B**) Expression of *Wnt2b* in mouse retina with single-cell transcriptomics. Dot plot of *Wnt2b* and its receptors (FZD4, FZD7, LRP5, and LRP6) gene expression (scaled) for different retinal cell types in P14 C57BL/6J mouse retinas. *Wnt2b* showed modest levels of expression in retina cells except for the cholinergic amacrine cell cluster. Data source: Study—P14 C57BL/6J mouse retinas (https://singlecell.broadinstitute.org/single_cell/study/SCP301, accessed on 27 January 2022) [27]. Scaling is relative to each gene’s expression across all cells of a cluster. Gene expression is scaled from zero to one (0.5 = the mean across all cells in the cluster file referenced for dot plotting). % expressing is the percent of cells that have one or more transcripts for the gene of interest. Dot size represents the percentage of cells expressing the specific gene, color indicates the relative expression level (red = higher expression).

**Figure 8 cells-12-01033-f008:**
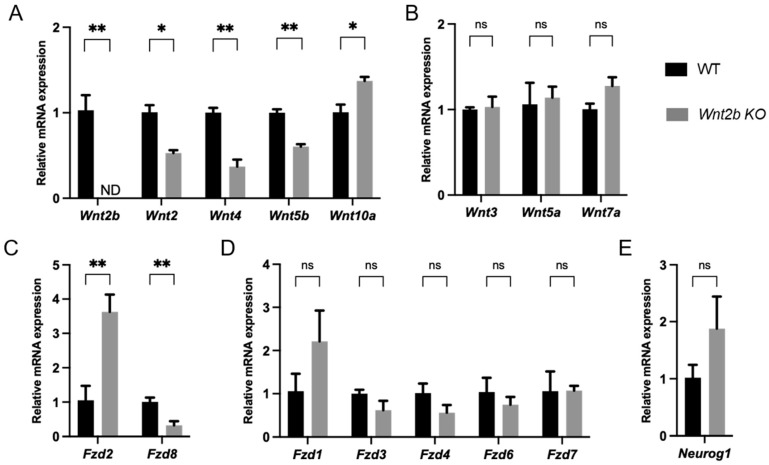
**Altered expression levels of Wnt ligands and frizzled (Fzd) receptors in *Wnt2b* KO retinas.** (**A**) Altered expression levels of *Wnt2*, *Wnt4*, *Wnt5b*, and *Wnt10a* mRNA in WT versus *Wnt2b* KO retinas from young adult mice (3–4-month-old). (**B**) Comparable levels of *Wnt3*, *Wnt5a*, and *Wnt7a* in WT versus Wnt2b KO retinas. (**C**) Altered expression levels of *Fzd2* and *Fzd8* mRNA in WT versus *Wnt2b* KO retinas. (**D**) Levels of *Fzd1*, *Fzd3*, *Fzd4*, *Fzd6*, and *Fzd7* were not significantly changed in WT versus *Wnt2b* KO retinas. (**E**) Expression levels of *neurogenin1* (*Neurog1*) showed an upregulated trend in KO retinas. n = 3–4 retinas /group. * *p* < 0.05, ** *p* < 0.01. ND: not detected. ns: not significant.

**Table 1 cells-12-01033-t001:** List of mouse primer sequences used for RT-qPCR.

Primer	Forward Sequence 5′-3′	Reverse Sequence 5′-3′
*Wnt2b*	GCCCGAGTGATCTGTGACAA	CACTCTCGGATCCATTCCCG
*Wnt2*	CTCGGTGGAATCTGGCTCTG	CACATTGTCACACATCACCCT
*Wnt3*	CCGCTTCTGTCTAGGGTCTG	GTAGAGAGTGCAGGCAAGGG
*Wnt4*	AGACGTGCGAGAAACTCAAAG	GGAACTGGTATTGGCACTCCT
*Wnt5a*	CAACTGGCAGGACTTTCTCAA	CATCTCCGATGCCGGAACT
*Wnt5b*	GGTTCCACTGGTGTTGCTTT	AGACTTTTGTGAGGCGGAGA
*Wnt7a*	CACTTGTGGTCTCAGGGGTT	GCATCTGAGTTTCACCAGCA
*Wnt10a*	GCTCAACGCCAACACAGTG	CGAAAACCTCGGCTGAAGATG
*Fzd1*	CAGCAGTACAACGGCGAAC	GTCCTCCTGATTCGTGTGGC
*Fzd2*	CATGCCCAACCTTCTTGGC	CAGCGGGTAGAACTGATGCAC
*Fzd3*	ATGGCTGTGAGCTGGATTGTC	GGCACATCCTCAAGGTTATAGGT
*Fzd4*	AGACGTGCGAACTCAAAG	GGAACTGGTATTGGCACTCCT
*Fzd6*	ATGGAAAGGTCCCCGTTTCTG	GGGAAGAACGTCATGTTGTAAGT
*Fzd7*	GCCACACGAACCAAGAGGAC	CGGGTGCGTACATAGAGCATAA
*Fzd8*	ATGGAGTGGGGTTACCTGTTG	CACCGTGATCTCTTGGCAC
*Neurog1*	AGTAGTCCCTCGGCTTCAGA	TATGGGATGAAACAGGGCGT
*Gapdh*	AACAGCAACTCCCACTCTTC	CCTGTTGCTGTAGCCGTATT

**Table 2 cells-12-01033-t002:** **Frequency analysis of ectopic photoreceptor phenotype observed** in H&E cross sections, organized by age and genotype. Values indicate the number of mice exhibiting phenotype/total number of mice examined.

Mouse Age	Genotype		
	WT or *Wnt2b* fl/fl *	*Wnt2b* Het	*Wnt2b* KO
**P7**	0/4	1/4	1/3
**P15**	2/6	2/3	1/4
**P30**	1/4	1/2	1/4
**2–5 mo**	1/6	4/5	3/6
**9 mo**	0/3	0/1	1/3
**Total**	4/23 (17%)	8/15 (53%)	7/20 (35%)

* Note: Fl/fl only: 3–4 mo, 0/4, total 0%. WT only: 2–5 mo, 1/2, total 4/19 = 21%.

## Data Availability

The paper contains all methods and data needed to evaluate the conclusions.

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
