# Peer review of "Ectopic Rod Photoreceptor Development in Mice with Genetic Deficiency of WNT2B"

_cells, 2023, doi:10.3390/cells12071033_

Round 1
Reviewer 1 Report
1. It will be good to validate the cre expression in the mouse retina.
2. Figure 2 should also include the images from the wt control
3. No n number for Figure 6D.
4. The paper left an unanswered question. What caused the phenotype? or how wnt2B KO caused the phenotype? Extra RNA sequencing experiment may provide some clues.
Author Response
Please see attached letter.

Reviewer 2 Report
Given the phenotype mutations in WNT2B show in humans it make sense to investigate the deletion of Wnt2b in mice.
1. Since acting via the canonical WNT-signaling pathway the question arises why the authors decided to use a general Cre-deleter line for the inactivation of Wnt2b? Why not using a retina-specific line like the RX-Cre or mRX-Cre transgenic mouse line to rule out systemic effects when deleting Wnt2b?
2. Despite the deletion of Wnt2b the authors show that the overall appearance of the eyes are normal, neither lens nor retinal vasculature seem to be effected by the gene deletion. They further analyze the eye histology. In Fig. 3 they show ectopic cells (identified by nuclear stain) in subretinal position. Labelling for rhodopsin indicates their rod photoreceptor origin. Some higher magnifications would be helpful to get an idea on the morphology of these ectopic cells.Are these cells rod precursors? Do they display normal rod morphology? With the images presented, this can not be judged. Also, ultra-structure (TEM) would be helpful to get an better idea of the morphology.
3. Another point to consider, would be the localization of this ectopic cells. Are the in the periphery of the eye or are they more evenly distributed? A schematic drawing indicating the distribution of these cells would be nice.
4. Fig. 4 : The images show pigment deposition in close proximity to the ectopic cells. Again, high resolution, high magnification images are missing. Are these cells what we are seeing? Is it cellular debris? Is this phenomenon also seen in WT- (Wnt2b fl/fl) and heterozygous littermates?
5. What is the explanation for the stronger phenotype in heterozygous mice in comparison to WT-littermates or Wnt2b-KO mice?
6. When looking at the Wnt2b expression profile in retinal cells and pigment epithelium by single-cell-transcriptomics the data are based on a C57BL/6J background. How do this data relate to the mixed genetic background used by the author? Is theire anything known about strain-specific expression of Wnt ligands and frizzled receptors?
Minor points:
Methods section 2.4 line 100: “Sections were air dried and briefly fixed…” How are the sections fixed? Please add details.
OCT: Optical coherence tomography with an N of 2 (mice) might not be meaningful, even if their would be a visible effect.
Author Response
Please see attached letter

Round 2
Reviewer 1 Report
The authors addressed all my concerns.
Author Response
Thank you.
Reviewer 2 Report
Overall, the quality of the presented paper improved by the changes made, still the morphology of the ectopic rods remains un-resolved and could have been addressed better.
The authors claim: “Higher magnification insets are now included in Fig. 3 to better visualize the ectopic cells (see below and updated Fig. 3).”
Nevertheless, it would have been good, if the author would have really presented higher magnifications of ectopic rods and not cropped-out and magnified regions of pre-existing pictures. What is the magnification factor?
“They don’t exhibit normal rod morphology, likely due in to lack of space for proper formation and placement of rod outer segment and axonal extension into outer plexiform layer, although it is difficult to visualize subcellular compartments from individual cell.”
As stated by the authors it will be difficult to visualize subcellular compartments (particularly with such low resolution and magnifications provided), however some more co-labeling experiments with markers other than rhodopsin, e.g., synaptic markers like SV2 or Ctbp2 could have provided insides in the overall morphology of the ectopic rods, e.g. whether this ectopic cells do not only exhibit the photoconversion compartment but also form synaptic terminals.
I agree with the authors, that performing EM analyses is challenging within 10 days of revision time, however performing co-immunolabeling experiments or at least taking higher-resolution images at higher magnification is feasible and cannot be considered as major new experiment.
